# Irregular Shifting of RF Driving Signal Phase to Overcome Dispersion Power Fading

**Febrizal Ujang** , **Teguh Firmansyah, Purnomo S. Priambodo and Gunawan Wibisono** *

Department of Electrical Engineering, Universitas Indonesia, Jawa Barat 16424, Indonesia;
febrizal@ui.ac.id (F.U.); teguh.firmansyah81@ui.ac.id (T.F.); p.s.priambodo@ieee.org (P.S.P.)
* Correspondence: gunawan@eng.ui.ac.id

**Abstract:** The main problem with the radio-over-fiber (RoF) link is the decrease in the recovered radio frequency (RF) power due to the chromatic dispersion of the fiber known as dispersion power fading. One of the methods for dealing with dispersion power fading is to use the optical single sideband (OSSB) modulation scheme. The OSSB modulation scheme can be generated by biasing the dual-drive Mach–Zehnder modulator (DD-MZM) to the quadrature bias point (QBP) and shifting the RF drive signal phase ($\theta$) by 90°, which is called the regular $\theta$. However, the OSSB modulation scheme only overcomes dispersion power fading well at the modulation index ($m$) < 0.2. This paper proposes an irregular $\theta$ method to overcome dispersion power fading at all $m$. There are two irregular $\theta$ for every $m$ used. The irregular $\theta$ managed to handle dispersion power fading better than OSSB modulation scheme did at every $m$. Specifically, the irregular $\theta$ could handle the dispersion power fading well at $m \leq 1$. In sum, the irregular $\theta$ could overcome the dispersion power fading at any RF frequency and optical wavelength without having to re-adjust the transmitter.

**Keywords:** radio over fiber; dispersion power fading; dual-drive Mach–Zehnder modulator (DD-MZM); irregular phase shift

## 1. Introduction

A radio-over-fiber (RoF) system transmits radio frequency (RF) ($X_{TX}(t)$) signals through optical fiber used to support wireless communication services. The $X_{TX}(t)$ in the RoF system is converted to an optical signal using an electro-optic (E/O) converter located at the central office (CO). The optical signal is later transmitted through a fiber link, and the RF signal is recovered using an opto-electric (O/E) converter positioned on the radio access point (RAP). The recovered RF signal ($X_{rec}(t)$) is then transmitted wirelessly from the RAP to mobile station (MS). It is possible to convert RF signal to the optical one by modulating the optical source directly or externally, both of which serve as intensity modulation (IM) or phase modulation (PM). The intensity modulation is used most commonly because it has a simple system. Recovering RF signal can be done by direct detection (DD) using a photodetector [1].

IM on the RoF system produces modulated light waves ($E_{TX}(t)$) with double-sideband spectrum (double-sideband optical modulation, henceforth ODSB). When the ODSB signal is transmitted through a fiber link, the chromatic dispersion of the fiber causes the sideband and optical carrier to propagate at different speeds. This leads the modulated signal at the receiver ($E_{RX}(t)$) to experience a phase difference between the sideband and optical carrier by $\varphi$. The proportion of $\varphi$ follows the length of the fiber ($L$), the frequency of the RF signal ($f_m$) and the wavelength ($\lambda_c$) used. The phase difference causes the O/E process to generate two identical RF signals but with a different phase of $2\varphi$, resulting in constructive and destructive interference on $X_{rec}(t)$. The destructive interference reduces the power of the recovered RF signal ($P_{rec}(t)$), which is known as dispersion power fading. If $\varphi = \pi$, massive decrease in power will

occur (deep fade). The proportion of power decrease is obtained by comparing the signal power with and without the fiber, something that is known as the carrier-to-noise ($C/N$) penalty [2].

There are two types of light sources commonly used in RoF links, namely optical frequency comb (OFC) and single-mode laser. Chromatic dispersion fiber will give different effects on each light source. The effect of chromatic dispersion on RoF links using OFC light sources can be overcome by several methods. In [3], power fading is overcome by utilizing chromatic dispersion. The effect of chromatic dispersion on RoF links using the mode-locked laser diodes (MLLD) can be reduced using an unbalanced Mach–Zehnder interferometer (UMZI) [4], whereas in [5], the effect of chromatic dispersion is reduced by adjusting the MLLD parameters appropriately.

The focus of this paper is to overcome dispersion power fading on RoF links using a single-mode laser as a light source. The most useful method for dealing with dispersion power fading is to use spools of dispersion compensation fiber (DCF) [6]. The dispersion of DCF is negative, so by inserting DCF into the ROF link, the average dispersion is close to zero. The weakness of this method is that DCF must always be adjusted to the $L$ used.

Dispersion power fading can also be overcome by carrier phase-shifted (CPS) method [7–12]. In this method, the optical carrier phase before transmission is arranged in such a way that $\varphi$ becomes zero after transmission. With this method, the $X_{rec}(t)$ is always in a constructive condition. However, the optical carrier phase before transmission should always be adjusted to the used $L$, $f_m$, and $\lambda_c$ since the proportion of $\varphi$ accords with $L$, $f_m$, and $\lambda_c$. In addition, successfully adjusting the optical carrier phase requires complex transmitter circuits.

Another feasible method is using the optical carrier-suppressed (OCS) modulation scheme [13–20]. With the OCS modulation, the optical carrier is removed so that $E_{TX}(t)$ consists only of upper and lower sidebands. The $X_{rec}(t)$ is thus generated only from the multiplication between the upper and lower sidebands, so no interference resulting in dispersion power fading is possible. The disadvantage of this method is that the $X_{rec}(t)$ frequency is twice the $X_{TX}(t)$ frequency, so the receiver has to do additional works to turn the $X_{rec}(t)$ frequency to its original frequency.

It is also possible to deal with dispersion power fading by using an optical single-sideband (OSSB) modulation scheme [13,15,16,19–43] that has a spectrum consisting only of optical carriers and one sideband (either upper or lower sideband). This modulation scheme allows the $X_{rec}(t)$ to have the same frequency as that of $X_{TX}(t)$, so it is not necessary for the receiver to adjust the $X_{rec}(t)$ frequency. The OSSB modulation scheme can be generated by biasing the dual-drive Mach–Zehnder modulator (DD-MZM) on the quadrature bias point (QBP) and at the phase difference of RF drive signal ($\theta$) = 90° [19,30,40,43,44]. The DD-MZM is an electro-optic ($E/O$) converter that is commonly used on RoF links. The OSSB generated using DD-MZM manages to effectively overcome dispersion power fading if the $X_{TX}(t)$ spectrum is made up only of optical carriers and sideband fundamentals without harmonics. Otherwise, the phase shift occurring in the harmonics may result in a decrease of the $X_{rec}(t)$ power. To produce $X_{TX}(t)$ without harmonics, the DD-MZM must be operated at a modulation index ($m$) < 0.2. Hence, this method cannot overcome dispersion power fading efficiently at $m \geq 0.2$.

Our previous study [45] has successfully demonstrated that modifying $\theta$ of DD-MZM biased on QBP can change the $X_{TX}(t)$ spectrum. Different spectrum will also produce a different level of dispersion power fading. This paper proposes, as an update, the use of irregular $\theta$ to overcome dispersion power fading. The irregular $\theta$ is a $\theta$ that produces a minimum level of dispersion power fading, which is measured using ($C/N$) deviation factor. To calculate the ($C/N$) deviation factor, it is necessary, at first, to model the recovered RF signal's power.

The complete contribution of this paper includes:

1.  A new method for dealing with dispersion power fading by using irregular $\theta$.
2.  Generating a simple RoF link with standard DD-MZM as an $E/O$ converter that can overcome dispersion power fading at all $m$. This link:

    (a)  Can be used in any $f_m$, $L$, and $\lambda_c$ without having to re-adjust the transmitter.

(b)　　Has $X_{rec}(t)$ set at the same frequency as that of $X_{TX}(t)$, thereby removing any additional work.

## 2. Principles of DD-MZM

The Mach–Zehnder modulator (MZM) is an external intensity modulator commonly used on optical fiber links. It has two variants, namely single drive (SD-MZM) and dual drive (DD-MZM). The latter has two signal driver ports and two bias ports (up and down) that can be controlled independently to produce a number of different forms of optical modulation. The general configuration of DD-MZM is shown in Figure 1 in which the path is distinguished by solid and dashed lines. The dashed lines represent the path passed by the electrical domain signal, while the solid one is passed by the optical domain signal. The continuous optical wave ($E_{in}(t)$) produced by the laser diode (LD) is modulated by the RF signal ($X_{TX}(t)$) to be transmitted later. The modulation is carried out by inserting $X_{TX}(t)$ into the upper and lower signal driver ports whose phases are differentiated using an electrical phase shifter (EPS) by $\theta$. The upper bias port is given a $V_{bias}$ voltage, while the lower one is given 0 V voltage. The modulator output optical field is expressed as $E_{TX}(t)$.

$E_{in}(t)$ and $X_{TX}(t)$ are expressed in the following formula:

$$E_{in}(t) = E_o e^{j2\pi f_c t} \tag{1}$$

$$X_{TX}(t) = V_m \cos 2\pi f_m t, \tag{2}$$

where $E_o$ represents continuous optical wave amplitude, $f_c$ the continuous optical wave frequency, $V_m$ the amplitude of the RF signal, and $f_m$ the frequency of the RF signal. As such, the upper ($V_{up}(t)$) and lower ($V_{down}(t)$) driving signals of DD-MZM are expressed:

$$V_{up}(t) = V_m \cos(2\pi f_m t) + V_{bias} \tag{3}$$

$$V_{down}(t) = V_m \cos(2\pi f_m t + \theta), \tag{4}$$

where $\theta$ is the phase of the electrical phase shifter (EPS). Assuming that the extinction ratio (ER) of DD-MZM is extremely high, the optical field generated from DD-MZM ($E_{TX}(t)$) can be approximated by the equation

$$E_{TX}(t) \approx \frac{1}{2} E_{in}(t) \left\{ e^{\left( j\pi \frac{V_{up}(t)}{V_\pi} \right)} + e^{\left( j\pi \frac{V_{down}(t)}{V_\pi} \right)} \right\}, \tag{5}$$

provided $V_\pi$ is the switching voltage of MZM. By inserting the Equations (3) and (4) into (5), the following formula is obtained

$$E_{TX}(t) \approx \frac{1}{2} E_{in}(t) \left\{ e^{\left( j\pi \frac{V_m}{V_\pi} \cos(2\pi f_m t) + j\pi \frac{V_{bias}}{V_\pi} \right)} + e^{\left( j\pi \frac{V_m}{V_\pi} \cos(2\pi f_m t + \theta) \right)} \right\}, \tag{6}$$

For simplicity, the Equation (6) can be stated as

$$\begin{aligned} E_{TX}(t) &= \tfrac{1}{2} E_{in}(t) \left\{ e^{\left( jm\cos(2\pi f_m t) + j\pi\gamma \right)} + e^{jm\cos(2\pi f_m t + \theta)} \right\} \\ E_{TX}(t) &= \tfrac{1}{2} E_{in}(t) \cdot \left\{ e^{jm\cos(2\pi f_m t)} e^{j\pi\gamma} + e^{jm\cos(2\pi f_m t + \theta)} \right\}, \end{aligned} \tag{7}$$

where $m = \pi \frac{V_m}{V_\pi}$ is the DD-MZM modulation index, and $\gamma = \frac{V_{bias}}{V_\pi}$ is the normalized bias voltage.

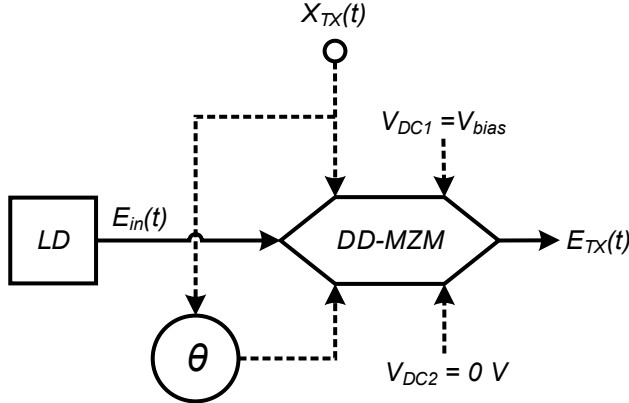

**Figure 1.** Schematic diagram of dual-drive Mach–Zehnder modulator (DD-MZM).

To facilitate the analysis, the Equation (7) is written as

$$E_{TX}(t) = \frac{1}{2}E_{in}(t)\left\{A \cdot e^{(j\pi\gamma)} + B\right\}, \tag{8}$$

with

$$A = e^{jm\cos(2\pi f_m t)} \tag{9}$$

$$B = e^{jm\cos(2\pi f_m t + \theta)}, \tag{10}$$

By implementing Jacobi–Anger expansion [46], where

$$e^{jm\cos x} = \sum_{n=-\infty}^{\infty} j^n J_n(m) e^{jnx} \tag{11}$$

$$e^{jm\sin x} = \sum_{n=-\infty}^{\infty} J_n(m) e^{jnx}, \tag{12}$$

the Equations (9) and (10) become

$$A = \sum_{n=-\infty}^{\infty} j^n \cdot J_n(m) \cdot e^{jn(2\pi f_m t)}, \tag{13}$$

$$B = \sum_{n=-\infty}^{\infty} j^n \cdot J_n(m) \cdot e^{jn(2\pi f_m t + \theta)}, \tag{14}$$

with $J_n(m)$ is the $n$th Bessel function of the first kind. Therefore, Equation (7) can be expressed as

$$E_{TX}(t) = \frac{1}{2}E_{in}(t)\left\{ \begin{array}{l} \sum\limits_{n=-\infty}^{\infty} j^n \cdot J_n(m) \cdot e^{jn(2\pi f_m t)} \cdot e^{j\pi\gamma} + \\ \sum\limits_{n=-\infty}^{\infty} j^n \cdot J_n(m) \cdot e^{jn(2\pi f_m t + \theta)} \end{array} \right\}$$

$$E_{TX}(t) = \frac{1}{2}E_{in}(t)\left\{ \sum_{n=-\infty}^{\infty} j^n \cdot J_n(m) \cdot \left(e^{jn(2\pi f_m t)} \cdot e^{j\pi\gamma} + e^{jn(2\pi f_m t)} \cdot e^{jn\theta}\right) \right\}$$

$$E_{TX}(t) = \frac{1}{2}E_{in}(t)\left\{ \sum_{n=-\infty}^{\infty} j^n \cdot \left(e^{j\pi\gamma} + e^{jn\theta}\right) \cdot J_n(m) \cdot e^{jn2\pi f_m t} \right\}, \tag{15}$$

By inserting (1) into (15), it is obtained

$$E_{TX}(t) = \tfrac{1}{2}E_o e^{j2\pi f_c t}\left\{ \sum_{n=-\infty}^{\infty} j^n \cdot \left(e^{j\pi\gamma} + e^{jn\theta}\right) \cdot J_n(m) \cdot e^{jn2\pi f_m t}\right\}$$

$$E_{TX}(t) = \tfrac{1}{2}E_o \left\{ \sum_{n=-\infty}^{\infty} j^n \cdot \left(e^{j\pi\gamma} + e^{jn\theta}\right) \cdot J_n(m) \cdot e^{j2\pi (f_c + nf_m) t}\right\}. \tag{16}$$

It is noticeable from Equation (16) that the optical field generated from DD-MZM consists of an optical carrier and sideband with infinite order. In accordance with the Bessel function, the formed sideband order is influenced by the $m$ used. The greater the $m$, the more the sideband orders formed.

By adjusting parameters $\gamma$ and $\theta$ in Equation (16), the following modulation schemes will be obtained:

1.  For $\theta = 180^o$ and $\gamma = \tfrac{1}{2}$ (QBP, which is $V_{bias} = \tfrac{1}{2}V_\pi$), an ODSB modulation scheme is produced. The spectrum of this modulated signal is shown in Figure 2a.
2.  For $\theta = 90^o$ and $\gamma = \tfrac{1}{2}$, an OSSB modulation scheme is produced. In this instance, only $(4n + 1)$ order lower sidebands are suppressed, where $n$ is an integer [19]. The spectrum can be seen in Figure 2b.

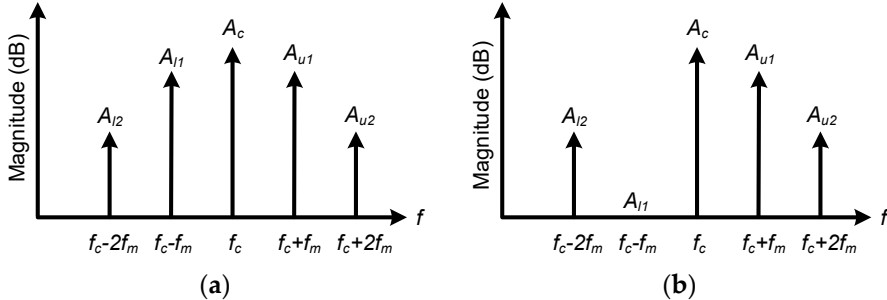

(**a**)　　　　　　　　　　　　　　　　(**b**)

**Figure 2.** The spectrum of a modulated optical signal. (**a**) DD-MZM as an optical double-sideband (ODSB) modulator and (**b**) OSSB modulator.

## 3. Modeling of Recovered RF Signal Power

To calculate the $(C/N)$ deviation factor, it is necessary to first model the recovered RF signal's power. Power modeling is carried out for the RoF link consisting of DD-MZM as an $E/O$ converter; optical fiber, which is dispersive with $H(f)$ response; and photodetector (PD), which is an O/E converter as shown in Figure 3.

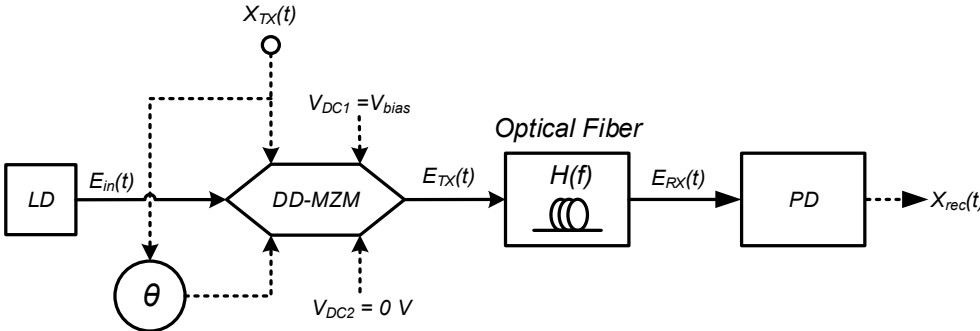

**Figure 3.** Radio-over-fiber (RoF) link with DD-MZM as electro-optic (*E/O*) converter.

To simplify the analysis, the Equation (16) can be rewritten as

$$E_{TX}(t) = A_c e^{j2\pi f_c t} + \sum_{n=1}^{\infty} \left\{ A_{ln} e^{j2\pi(f_c - n f_m)t} + A_{un} e^{j2\pi(f_c + n f_m)t} \right\}, \tag{17}$$

where

$$\begin{aligned}
A_c &= \tfrac{1}{2} E_o \left( e^{j\pi\gamma} + 1 \right) \cdot J_0(m) \\
A_{ln} &= \tfrac{1}{2} E_o j^n \left( e^{j\pi\gamma} + e^{-jn\theta} \right) \cdot J_n(m) \\
A_{un} &= \tfrac{1}{2} E_o j^n \left( e^{j\pi\gamma} + e^{jn\theta} \right) \cdot J_n(m).
\end{aligned}$$

The function of dispersive fiber link transfer is obtained from [47]

$$H(f) = e^{\frac{jDL\lambda_c^2(f-f_c)^2}{c}}, \tag{18}$$

where $D$ is the chromatic dispersion in ps/(nm km), $\lambda_c$ the optical wavelength, $f$ the frequency offset of the optical carrier, $c$ the speed of light in vacuum, and $L$ the fiber length in km. When $E_{TX}(t)$ is transmitted through a dispersive link, a phase difference occurs between the first-order sideband and the optical carrier

$$\phi = \frac{DL\lambda_c^2 f_m^2}{c}, \tag{19}$$

while the phase difference between the optical carrier and any sideband is the square of the frequency range $(\pm n f_m)$, which is given by [48]

$$\phi_n = n^2 \phi, \tag{20}$$

Thus, the optical field that arrives at the receiver end $(E_{RX}(t))$ is formulated below:

$$E_{RX}(t) = A_c e^{j2\pi f_c t} + \sum_{n=1}^{\infty} \left\{ A_{ln} e^{j2\pi(f_c - n f_m)t} + A_{un} e^{j2\pi(f_c + n f_m)t} \right\} \cdot e^{jn^2\phi}, \tag{21}$$

At the receiver end, the $E_{RX}(t)$ is detected using a photodetector which is a squared-envelope operator, given by [2]

$$\left| E_{RX}(t) \right|^2 = E_{RX}(t) \cdot E_{RX}^*(t), \tag{22}$$

by simply taking the $f_m$ term, it is obtained

$$\begin{aligned}
\left| E_{RX}(t) \right|^2 &= e^{j2\pi f_m t} \left\{ A_c A_{l1}^* e^{-j\phi} + A_c^* A_{u1} e^{j\phi} + A_{l1} A_{l2}^* e^{-j3\phi} + A_{u1}^* A_{u2} e^{j3\phi} + A_{l2} A_{l3}^* e^{-j5\phi} + A_{u2}^* A_{u3} e^{j5\phi} + \ldots \right\} \\
\left| E_{RX}(t) \right|^2 &= e^{j2\pi f_m t} \left\{ \sum_{n=0}^{\infty} \left( A_{ln} A_{l(n+1)}^* e^{-j(2n+1)\phi} + A_{un}^* A_{u(n+1)} e^{j(2n+1)\phi} \right) \right\},
\end{aligned} \tag{23}$$

where $A_{l0} = A_{u0} = A_c$.

The real portion of Equation (23) is equivalent to the photodetector's output current and is equal to $X_{rec}(t)$ [47], so

$$X_{rec}(t) \approx \left\{ \sum_{n=0}^{\infty} \left( A_{ln} A_{l(n+1)}^* e^{-j(2n+1)\phi} + A_{un}^* A_{u(n+1)} e^{j(2n+1)\phi} \right) \right\} \cos 2\pi f_m t. \tag{24}$$

The square of the amplitude term in the Equation (24) is the $X_{rec}(t)$ power [2], so the formula below is used to measure the power of the recovered RF signal as a function of $L$

$$P_{rec}(L) = \left\{ \sum_{n=0}^{\infty} \left( A_{ln} A_{l(n+1)}^* e^{-j(2n+1)\phi} + A_{un}^* A_{u(n+1)} e^{j(2n+1)\phi} \right) \right\}^2, \tag{25}$$

with

$$\phi = \frac{DL\lambda_c^2 f_m^2}{c},$$

$$A_{ln}^* = \tfrac{1}{2}E_o(-j^n)\left(e^{-j\pi\gamma} + e^{jn\theta}\right)\cdot J_n(m),$$

and

$$A_{un}^* = \frac{1}{2}E_o(-j^n)\left(e^{-j\pi\gamma} + e^{-jn\theta}\right)\cdot J_n(m).$$

The recovered RF signal is obtained from the multiplication of $n$th order lower sideband with $(n + 1)$th order conjugated lower sideband plus the multiplication of $n$th order conjugated upper sideband with $(n + 1)$th order upper sideband, where $n$ is an integer. The recovered RF signal power is obtained by squaring the amplitude of the signal.

The extent to which the power of the recovered RF signal decreases due to fiber dispersion is known as a $(C/N)$ penalty, which is calculated by comparing $P_{rec}(L)$ with and without fiber transmission. The calculation is mathematically stated as [2]

$$(C/N)_{penalty} = 10\log\left|\frac{P_{rec}(L)_{with fiber}}{P_{rec}(L)_{without fiber}}\right|. \tag{26}$$

This paper puts forward the idea of using $(C/N)$ deviation factor to measure the dispersion power fading level satisfying the following equation:

$$(C/N)_{deviation-factor} = \sqrt{\sum_{i=0}^{N}\frac{\left((C/N)_{penalty(i)} - \overline{(C/N)_{penalty}}\right)^2}{N}}. \tag{27}$$

Note: The $(C/N)_{penalty(i)}$ is the i-sample of $(C/N)$ penalty, $\overline{(C/N)_{penalty}}$ is the average sample of $(C/N)$ penalty, and $N$ is the number of samples. The lower the $(C/N)$ deviation factor, the smaller the effect of dispersion on $P_{rec}(L)$. The minimum sample required for this calculation is one deep fade cycle, as shown in Figure 4a.

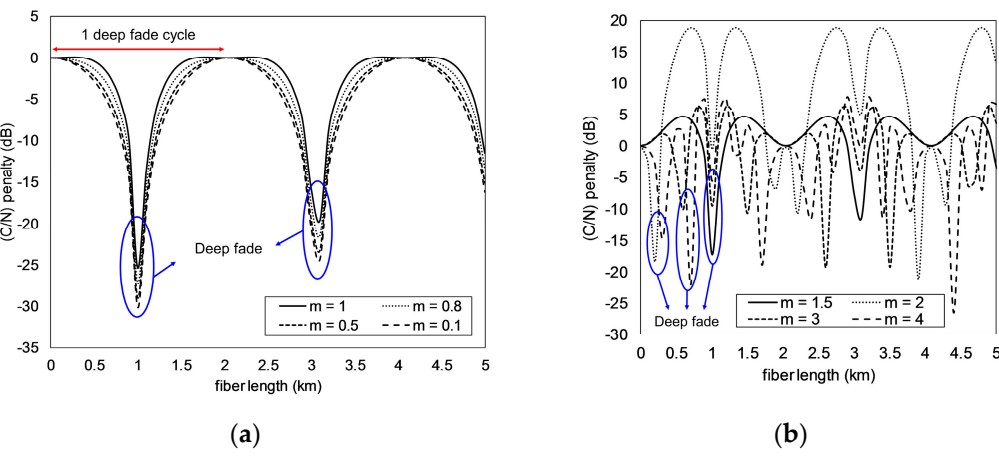

**Figure 4.** Carrier-to-noise $(C/N)$ penalty on the RoF link with ODSB modulation (**a**) $m \le 1$ (**b**) $m > 1$.

### 3.1. (C/N) Penalty on ODSB

The modulation scheme commonly used on RoF links is optical double sideband (ODSB). The RoF link with DD-MZM as *E/O* converter will produce an ODSB modulation scheme if DD-MZM is biased to QBP and the degree of RF phase shifting ($\theta$) equals 180°. The calculation of $(C/N)$ penalty on the RoF link with ODSB modulation follows (26) by calculating $P_{rec}(L)$ from the Equation (25), where $P_{rec}(L)$ without fiber = $P_{rec}(0)$, and $(C/N)$ penalty (dB) = $P_{rec}(L)$ (dBm) − $P_{rec}(0)$ (dBm). The parameters used in this calculation are $\lambda_c$ = 1550 nm with D = 17 ps/(nm.km), and $f_m$ = 60 GHz. The range of fiber length

$L$ is 0 to 5 km with a step of 0.1 km. The calculations are made at $m$ = 0.1, 0.5, 0.8, 1, 1.5, 2, 3, and 4. According to Bessel function, $J_n(4)$ is still significant until $n$ = 10. Thus, the considered sideband in this calculation is that reaching until the 10th order ($n$ = 10). The ($C/N$) penalty curve as the result of the calculation is shown in Figure 4.

The vertical axis in Figure 4 expresses the ($C/N$) penalty in dB, and the horizontal axis expresses the fiber length in km. Figure 4a shows the ($C/N$) penalty curve at $m \leq 1$, while Figure 4b is the ($C/N$) penalty curve at $m > 1$. It is obvious from Figure 4a that the RoF link with ODSB modulation at all $m$ experiences deep fade (large power loss) at $L$ = 1 and 3.1 km. Based on the Equation (26), the value of ($C/N$) deviation factor of RoF link with ODSB modulation is 6.8 at $m$ = 0.1, 6.5 at $m$ = 0.5, 6.0 at $m$ = 0.8, and 5.5 at $m$ = 1. The values indicate that the smaller the $m$, the larger the ($C/N$) deviation factor of RoF link with ODSB modulation. In other words, smaller $m$ values make the RoF link with ODSB modulation increasingly affected by dispersion. As shown in Figure 4b, the RoF link at $m > 1$ experiences deep fade at the various length of the fiber and has an irregular pattern. The same case happens with the ($C/N$) deviation factor at $m > 1$, which is 4.1 at $m$ = 1.5, 10.5 at $m$ = 2, 5.6 at $m$ = 3, and 7.4 at $m$ = 4.

To find out the effect of dispersion on the RoF link at different $f_m$, the researchers did a calculation of ($C/N$) penalty with $m$ = 1, $\lambda_c$ = 1550 nm, D = 17 ps/(nm km) at $L$ = 1 km to 5 km with a step of 0.1 km. The $f_m$ was varied at 30, 40, 50, 60, and 70 GHz. The results are shown in Figure 5a. On the other hand, to establish the effect of dispersion to the RoF link at different $\lambda_c$, the researchers made a similar ($C/N$) penalty calculation at $m$ = 1, $f_m$ = 60 GHz, at $L$ = 1 km to 5 km with a step of 0.1 km. The $\lambda_c$ was also varied at 1540 nm ($D$ = 16 ps/(nm km)), 1550 nm ($D$ = 17 ps/(nm km)), 1560 nm ($D$ = 17.5 ps/(nm km)), and 1570 nm ($D$ = 18 ps/(nm km)), and the results of the calculation are given in Figure 5b. On the RoF link with variation $f_m$, the bigger the $f_m$ used, the closer the distance of the deep fade. Likewise, on the RoF link with variation $\lambda_c$, bigger $\lambda_c$ results in a closer distance of the deep fade.

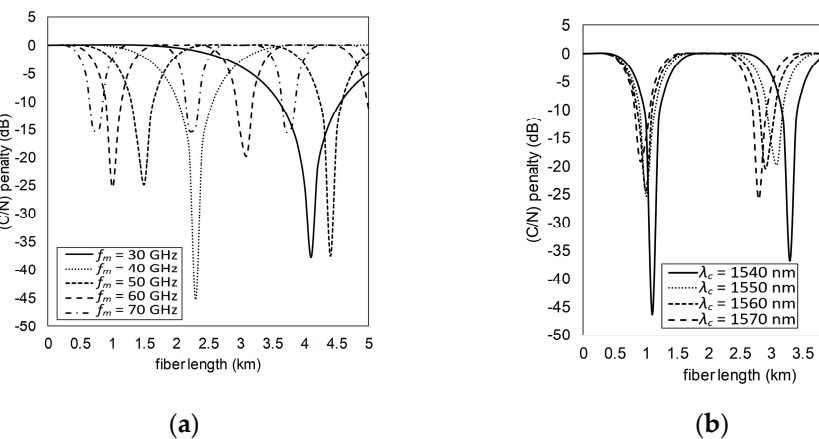

(**a**)                                                             (**b**)

**Figure 5.** The ($C/N$) penalty on the RoF link with ODSB modulation (**a**) variation $f_m$, (**b**) variation $\lambda_c$.

### 3.2. (C/N) Penalty on OSSB

One method used to manage the ($C/N$) penalty is the OSSB modulation scheme. The RoF link with DD-MZM as the *E/O* converter will produce an OSSB modulation scheme if DD-MZM is biased to QBP and $\theta$ is set to 90° [19,30,40,43,44]. To see the ($C/N$) penalty of RoF link with OSSB modulation, the same calculation as that on ODSB is done but with $\theta$ set to 90°. The results of ($C/N$) penalty from this calculation are shown in Figure 6.

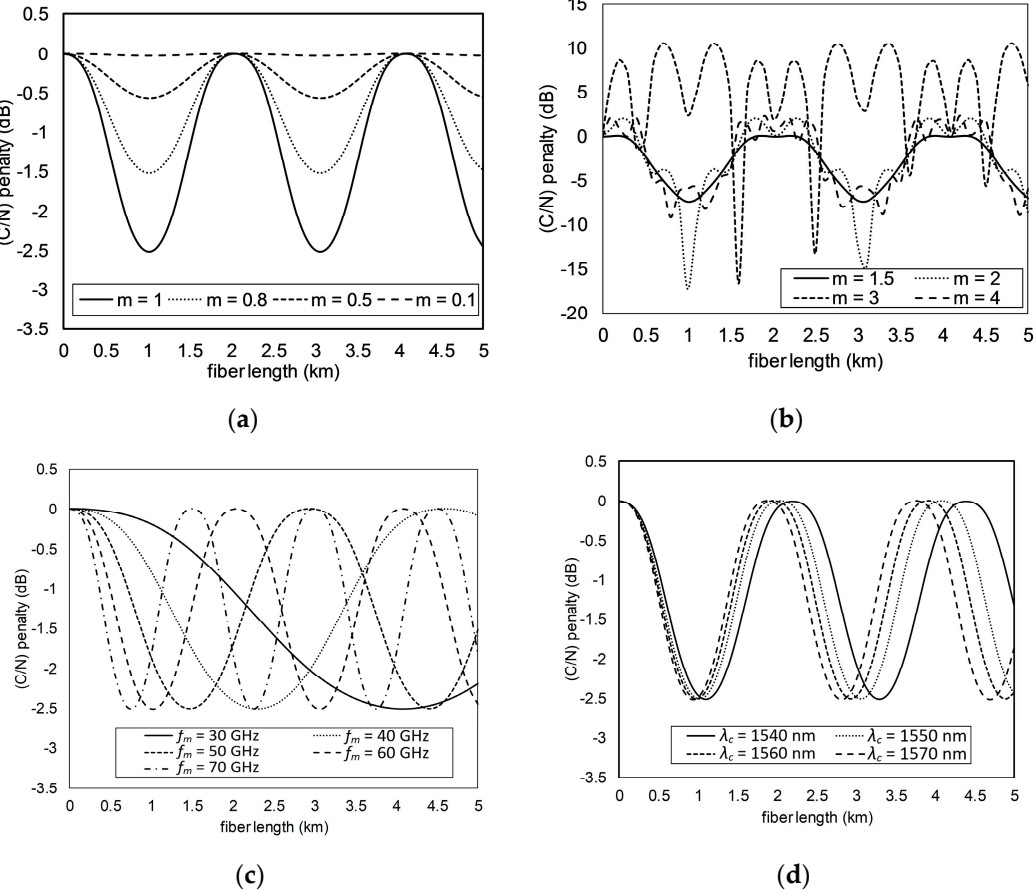

**Figure 6.** (*C/N*) penalty on the RoF link with OSSB modulation, (**a**) $m \leq 1$, (**b**) $m > 1$, (**c**) variation $f_m$ and (**d**) variation $\lambda_c$.

Figure 6a displays (*C/N*) penalty curve on the RoF link with OSSB modulation at $m \leq 1$. At this value of *m*, no deep fade was recorded, unlike that with ODSB modulation, but there remained a decrease of power by 0.6 dB at $m = 0.5$, 1.5 dB at $m = 0.8$, and 2.5 dB at $m = 1$. In other words, the greater the *m* used, the greater the decrease of power. The power reduction did not occur at $m = 0.1$, which means that the OSSB modulation scheme can only overcome the (*C/N*) penalty at $m = 0.1$. The value of (*C/N*) deviation factor of RoF link with OSSB modulation is 0.9 at $m = 1$, 0.6 at $m = 0.8$, 0.2 at $m = 0.5$, and 0.0 at $m = 0.1$. Hence, the RoF link with the OSSB modulation scheme is not, by implication, affected by the dispersion at $m = 0.1$—the larger the *m*, the greater the effect of dispersion on the recovered RF signal power. What the curve of (*C/N*) penalty and the (*C/N*) deviation factor show is that the OSSB modulation scheme cannot effectively overcome dispersion power fading at $m > 0.1$.

The image of (*C/N*) penalty curve on the RoF link with OSSB modulation at $m > 1$ is given in Figure 6b. It is visible at this level of *m* that the OSSB modulation scheme is no longer capable of overcoming dispersion power fading, which is marked by the significant loss of power at every use. Figure 6c,d is (*C/N*) penalty curve on the RoF link with OSSB modulation at various $f_m$ and $\lambda_c$ respectively. The curve was obtained from calculations with $m = 1$, $\lambda_c = 1550$ nm, and D = 17 ps/(nm km). Both figures illustrate the loss of power by 2.5 dB at different fiber length for different $f_m$ and $\lambda_c$. This implies that the OSSB modulation scheme is also unable to overcome dispersion power fading effectively at other $f_m$ and $\lambda_c$ for $m > 0.1$.

## 4. Irregular Phase Shifted

Our previous research [45] proved that altering $\theta$ on the RoF link with DD-MZM biased in QBP causes the $X_{TX}(t)$ spectrum to change as well. The $X_{TX}(t)$ with a different spectrum will result in a different $(C/N)$ deviation factor. The proper shape of the spectrum will produce a small $(C/N)$ deviation factor value, so choosing the right irregular $\theta$ will produce a minimum $(C/N)$ deviation factor value.

The search for irregular $\theta$ is performed for $\gamma = \frac{1}{2}$ with the following steps:

(a) Calculate $P_{rec}(L)$ using (24) with $n = 10$, $m = 0.1$, $\theta = 0$ rad, $\lambda_c = 1550$ nm (D = 17 ps/(nm km)), and $f_m = 60$ GHz. The $P_{rec}(L)$ is calculated at $0 \leq L \leq 5$ km with step 0.1 km.
(b) From the obtained $P_{rec}(L)$ in a), calculate the $(C/N)$ penalty using (26).
(c) Calculate the $(C/N)$ deviation factor using (27) of all $(C/N)$ penalties in b).
(d) Repeat steps a) to c) for the value $0° \leq \theta \leq 360°$ with step $1°$.
(e) Find $\theta$ in step d) which produces the smallest $(C/N)$ deviation factor.
(f) Repeat steps a) to e) for $0.1 \leq m \leq 4$ with step 0.1. The value of $m$ is limited to 4 since only in this condition can the sidebands of >10 order be ignored.

The curve for the search of irregular $\theta$ is shown in Figure 7. The vertical axis represents the $(C/N)$ deviation factor, and the horizontal one represents the $\theta$ value, which varies from 0 to $360°$. The $\theta$ producing the smallest $(C/N)$ deviation factor is chosen as the irregular $\theta$. For each $m$, there are two irregular $\theta$s. The value of irregular $\theta$ I is $90°$ at $m = 0.1$, $86°$ at $m = 0.5$, $78°$ at $m = 0.8$, and $69°$ at $m = 1$. Meanwhile, the value of irregular $\theta$ II is $270°$ at $m = 0.1$, $274°$ at $m = 0.5$, $282°$ at $m = 0.8$, and $291°$ at $m = 1$. The value of irregular $\theta$ II = $360°$ − irregular $\theta$ I. The irregular $\theta$ for another $m$ is given in Table 1.

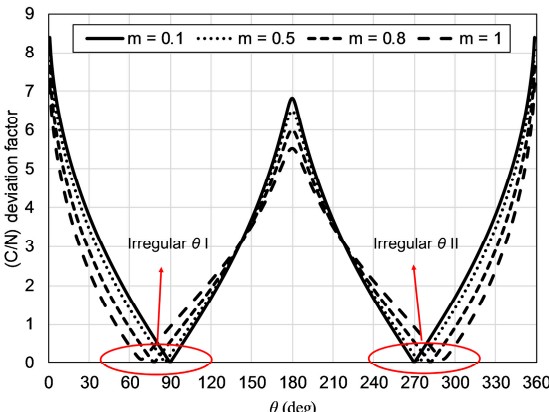

**Figure 7.** $(C/N)$ deviation factor as a function of $\theta$.

To clarify whether the irregular $\theta$ is periodical, the researchers investigated it at the range of $360°$ $\leq \theta \leq 720°$, and the results are presented in Figure 8a. For $m = 1$, the irregular $\theta$ I is $429°$ or $69° + 360°$, which means it is periodical. The irregular $\theta$ was also investigated at $f_m = 40$ GHz. The results of the calculation, as shown in Figure 8b, conclude that at $m = 1$, the irregular $\theta$ I amounts to $69°$. In other words, the irregular $\theta$ is the same with the same $m$ despite different $f_m$.

**Table 1.** Irregular θ for *m* = 0.1–4.

| m | Irregular θ I (Degrees) | Irregular θ II (Degrees) | m | Irregular θ I (Degrees) | Irregular θ II (Degrees) |
|---|---|---|---|---|---|
| 0.1 | 90 | 270 | 2.1 | 88 | 272 |
| 0.2 | 89 | 271 | 2.2 | 85 | 275 |
| 0.3 | 89 | 271 | 2.3 | 79 | 281 |
| 0.4 | 88 | 272 | 2.4 | 73 | 287 |
| 0.5 | 86 | 274 | 2.5 | 68 | 292 |
| 0.6 | 84 | 276 | 2.6 | 65 | 295 |
| 0.7 | 81 | 279 | 2.7 | 62 | 298 |
| 0.8 | 78 | 282 | 2.8 | 60 | 300 |
| 0.9 | 74 | 286 | 2.9 | 59 | 301 |
| 1.0 | 69 | 291 | 3.0 | 58 | 302 |
| 1.1 | 63 | 297 | 3.1 | 18 | 342 |
| 1.2 | 56 | 304 | 3.2 | 15 | 345 |
| 1.3 | 50 | 310 | 3.3 | 105 | 255 |
| 1.4 | 44 | 316 | 3.4 | 107 | 256 |
| 1.5 | 38 | 322 | 3.5 | 108 | 252 |
| 1.6 | 32 | 328 | 3.6 | 107 | 253 |
| 1.7 | 28 | 332 | 3.7 | 105 | 255 |
| 1.8 | 145 | 215 | 3.8 | 103 | 257 |
| 1.9 | 142 | 218 | 3.9 | 47 | 313 |
| 2.0 | 86 | 274 | 4.0 | 45 | 315 |

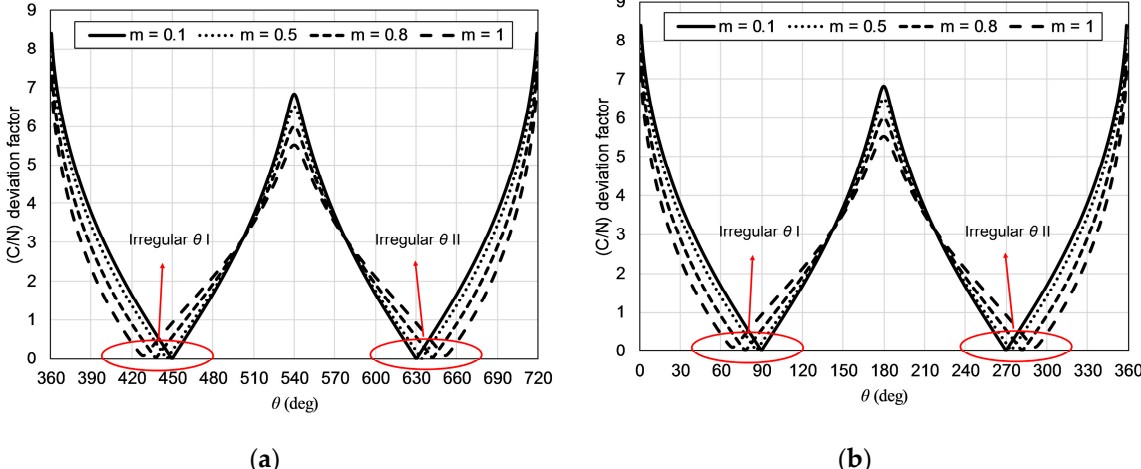

**Figure 8.** (*C*/*N*) deviation factor (**a**) 360° ≤ θ ≤ 720° (**b**) at $f_m$ = 40 GHz.

To see the proportion of decrease in (*C*/*N*) deviation factor before and after the use of irregular θ, a comparison of (*C*/*N*) deviation factor on the RoF link with ODSB, OSSB, and irregular θ modulation was performed. The calculation is performed using $\lambda_c$ = 1550 nm (*D* = 17 ps/(nm km)), and $f_m$ = 60 GHz at 0.1 ≤ *m* ≤ 4. The results of this calculation are shown in Figure 9.

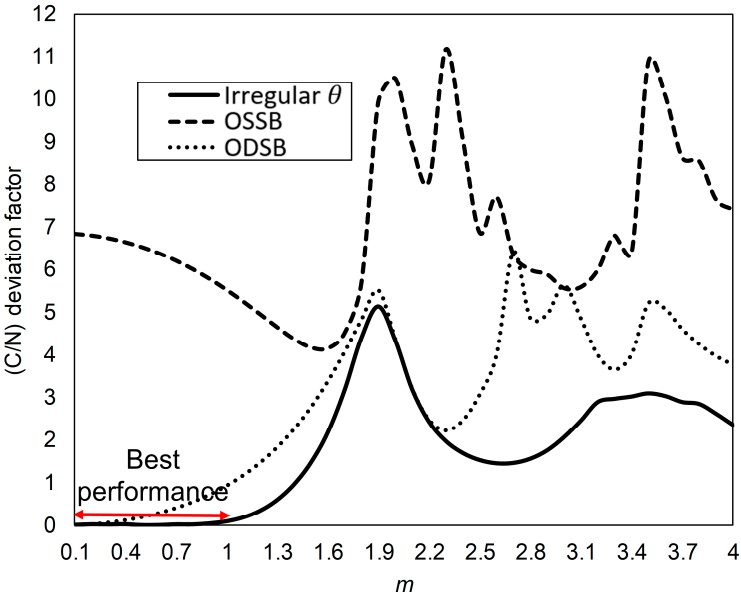

**Figure 9.** The (*C/N*) deviation factor of RoF link with ODSB, OSSB modulation schemes, and irregular *θ*.

The vertical axis stands for the (*C/N*) deviation factor, and the horizontal axis represents *m*. It is visible that the value of the (*C/N*) deviation factor of RoF link with ODSB modulation is always greater than those of OSSB modulation and irregular *θ*. The line of (*C/N*) deviation factor of ODSB at 0.1 < *m* < 1.6 decreased before fluctuating at *m* > 1.6. In contrast, the value of (*C/N*) deviation factor for OSSB and irregular *θ* increased at 0.1 < *m* < 1.9 and then fluctuated at *m* > 1.9. The (*C/N*) deviation factor for irregular *θ* is always smaller than OSSB. These findings suggest that the use of irregular *θ* is able to handle dispersion power fading better than OSSB does on all *m*. The RoF link with irregular *θ* has the (*C/N*) deviation factor < 0.1 at *m* ≤ 1. This means the irregular *θ* can cope with dispersion power fading well at *m* ≤ 1.

To see the performance of irregular *θ* in handling dispersion power fading, a calculation of the (*C/N*) penalty at *m* = 1 for RoF link calculation with irregular *θ* I = 69°, irregular *θ* II = 291°, ODSB modulation, and OSSB modulation was performed. The calculation was performed with $\lambda_c$ = 1550 nm (D = 17 ps/(nm.km)), and $f_m$ = 60 GHz. To validate these results, a comparison of calculation results of (*C/N*) penalty with the simulation results using Optisystem software was done. The simulation circuit is given in Figure 10.

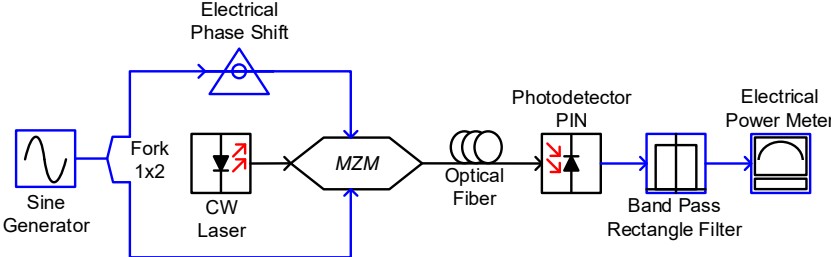

**Figure 10.** The structure of RoF link simulation using Optisystem software.

The structure of the RoF link consists of sine generator, fork 1x2, electrical phase shift, CW laser, MZM, optical fiber, photodetector PIN, band pass rectangle filter, and electrical power meter. The sine generator works to generate pure RF signals. In this simulation, the frequency of sine generator is set to 60 GHz. Because the switching voltage $V_\pi$ used in the simulation is 4 V, the voltage of sine generator $V_m$ is set to 1.274 V to obtain *m* = 1. The output of the sine generator is then duplicated using fork 1x2. The first fork output is inserted into the electrical phase shift and utilized as an MZM

top driver, while the second fork output is directly used as an MZM bottom driver. The electrical phase shift is used to differentiate the phase between the first RF signal and the second fork output by $\theta$. The type of MZM used is LiNb-MZM. To produce $\gamma = \frac{1}{2}$, MZM is set with the parameters as outlined in Table 2. MZM optical inputs are CW laser which is set to $\lambda_c$ = 1550 nm, power = 0 dBm, and line width = 10 MHz. The output of MZM is then transmitted through single-mode optical fiber. The optical fiber is configured with parameters in Table 3. In this simulation, the effect of fiber attenuation was ignored. At the receiver, the optical signal is detected using PIN photodetector under the parameter of responsivity (1 A/W) and dark current (10 nA). Because the output of photodetector consists of an electric signal with frequencies of 0, 60, 120 GHz, etc., it was filtered by means of a band pass rectangle filter. To obtain an RF signal at 60 GHz, the parameter filter was used by frequency of 60 GHz, bandwidth of 10 MHz, the insertion loss of 0 dB and a depth of 100 dB. The power of the recovered RF signal was measured using the electrical power meter. The simulation was performed in three scenarios, with the first one being for the RoF link with ODSB modulation scheme. To produce the ODSB modulation scheme, the electrical phase shift is set to $\theta = 180°$. The second scenario is to set $\theta = 90°$ to produce OSSB modulation scheme, while the third scenario is for the RoF link with the irregular $\theta$. Since $m$ in this simulation is 1, the electrical phase shift is set to $\theta = 69$ and $291°$. The measurements of power for each scenario were performed for 0 to 5 km fiber lengths with a step of 0.1 km.

**Table 2.** Setting parameter of LiNb-MZM.

| Parameter | Value | Units |
|---|---|---|
| Extinction ratio | 20 | dB |
| Switching bias voltage | 4 | V |
| Switching radio frequency (RF) voltage | 4 | V |
| Insertion loss | 0 | dB |
| Normalize electrical signal | unchecked | - |
| Bias voltage1 | 0 | V |
| Bias voltage2 | 2 | V |

**Table 3.** Setting parameter of optical fiber.

| Parameter | Value | Units |
|---|---|---|
| User-defined reference wavelength | Checked | - |
| Reference wavelength | 1550 | nm |
| Length | 0–5 | km |
| Attenuation effect | Unchecked | - |
| Group velocity dispersion | Checked | - |
| Third-order dispersion | Unchecked | - |
| Frequency domain parameter | Unchecked | - |
| Dispersion | 17 | ps/nm/km |

The ($C/N$) penalty curve for calculation and simulation results is shown in Figure 11. The curve for the results of both the calculation and simulation from the RoF link with ODSB modulation are the same in pattern. The same pattern of the curve also applied to OSSB modulation and irregular $\theta$ ($\theta = 69°$ and $291°$). This means that the mathematical model used to calculate the recovered RF signal power is correct. On the RoF link with the ODSB modulation scheme, a deep fade occurs at $L = 1$ and 3.1 km. The deep fade did not occur in the RoF link with OSSB modulation scheme, but there is a reduction in power of about 2.5 dB at the same $L$. Meanwhile, the power loss on the RoF link with irregular $\theta = 69°$ and irregular $\theta = 291°$ is extremely small. The ($C/N$) deviation factor of the RoF link with irregular $\theta = 69°$ and irregular $\theta = 291°$ is 0.1, with OSSB modulation 0.9, and ODSB modulation 5.5. Therefore, irregular $\theta$ could reduce the ($C/N$) deviation factor by 5.4, while OSSB reduced by 4.6. This suggests the irregular $\theta$ handles dispersion power fading better than OSSB does. Irregular $\theta = 69°$

and irregular $\theta = 291°$ have the same pattern of the curve and performance in handling dispersion power fading.

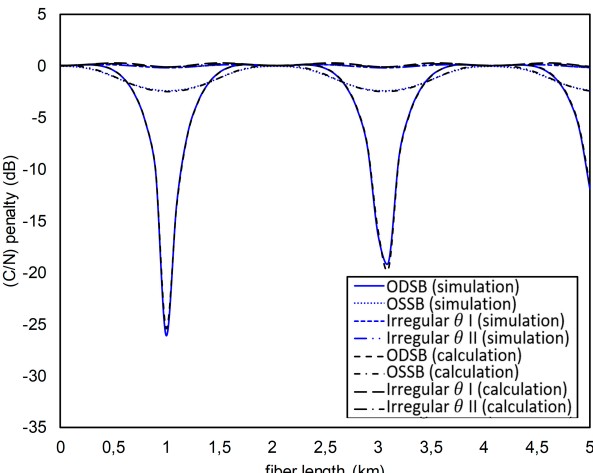

**Figure 11.** (*C/N*) penalty of RoF link with ODSB, OSSB modulation, and irregular $\theta$: The results *of* calculation and simulation.

The calculation of (*C/N*) penalty was also performed on the RoF link with $f_m = 30$ and $40$ GHz at $m = 1$ to test whether the use of irregular $\theta$ can successfully handle the dispersion power fading of RoF link at different $f_m$. The calculation is done with $\lambda_c = 1550$ nm ($D = 17$ ps/(nm km)). The curve for the results of this calculation is shown in Figure 12.

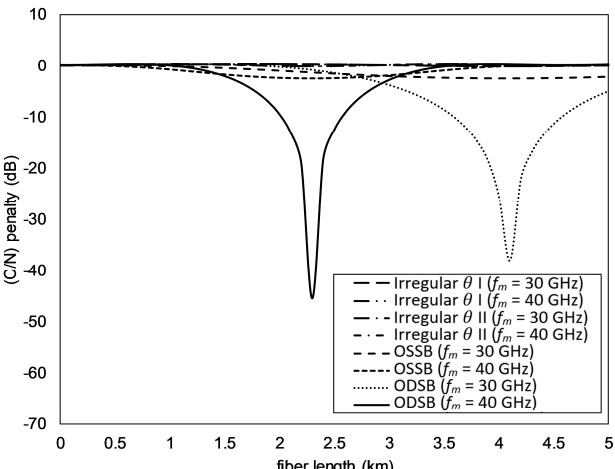

**Figure 12.** (*C/N*) penalty of RoF link with ODSB, OSSB modulation, and irregular $\theta$ for $f_m = 30$ and $40$ GHz.

It is obvious from Figure 12 that the ODSB modulation of RoF link with $f_m = 30$ GHz experienced a deep fade at $L = 4.1$ km, while that with $f_m = 40$ GHz experienced it at $L = 2.3$ km. No deep fade was recorded on the RoF link with OSSB modulation, but there remained a power reduction of 2.5 dB at the same $L$. The reduction on the irregular $\theta$ of RoF link was incredibly small. The (*C/N*) deviation factor with ODSB was 7.8 at $f_m = 30$ GHz and 7.5 at $f_m = 40$ GHz. On the other hand, the (*C/N*) deviation factor of OSSB was 1.0 at $f_m = 30$ GHz and 0.9 at $f_m = 40$ GHz, while that of irregular $\theta$ was 0.1 at $f_m = 30$ GHz and 0.1 at $f_m = 40$ GHz. The figures suggest that the irregular $\theta$ handles dispersion power fading better than OSSB in every $f_m$.

To find out whether the irregular $\theta$ can overcome dispersion power fading at another $\lambda_c$, the researcher conducted a (*C/N*) penalty calculation on the RoF link with $\lambda_c = 1540$ nm ($D = 16$ ps/(nm

km)) and $\lambda_c$ = 1570 nm (*D* = 18 ps/(nm km)) at *m* = 1. Figure 13 portrays the curve for the results of this calculation.

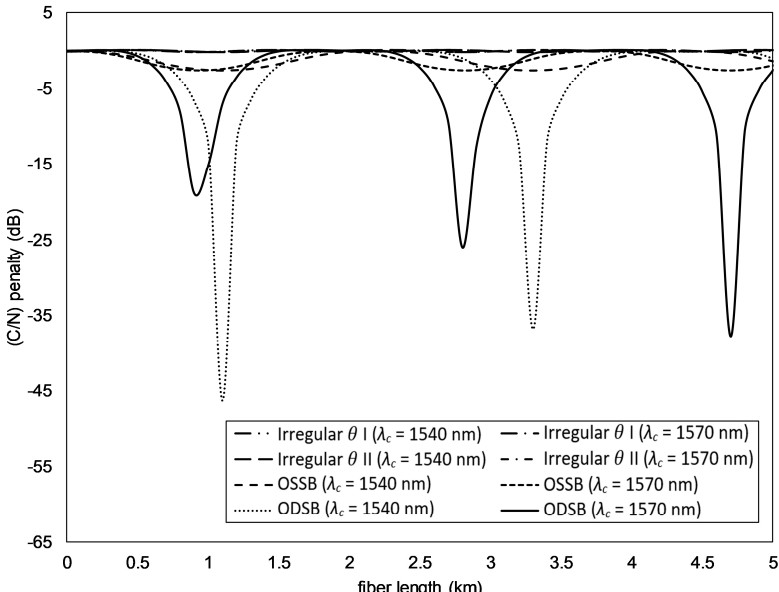

**Figure 13.** The (*C/N*) penalty for RoF link with ODSB, OSSB, and irregular $\theta$ modulation at $\lambda_c$ = 1540 nm and 1570 nm.

It was found that the RoF link of ODSB modulation with $\lambda_c$ = 1570 nm experienced a deep fade at *L* = 0.9, 2.8 and 4.8 km, while that with $\lambda_c$ = 1540 nm experienced the fade at *L* = 1.1 and 3.3 km. The deep fade did not occur in the RoF link of OSSB or irregular $\theta$, but a substantial power reduction of 2.5 dB occurred with OSSB modulation, whereas the reduction with the irregular $\theta$ was exceptionally small. The (*C/N*) deviation factor of RoF link with ODSB modulation was 8.6, OSSB modulation 0.9, and irregular $\theta$ 0.1, all at $\lambda_c$ = 1540 nm. The (*C/N*) deviation factor with $\lambda_c$ = 1570 nm for ODSB modulation was 7.3, OSSB modulation 0.9, and irregular $\theta$ 0.1. These numbers also imply that the use of irregular $\theta$ overcomes dispersion power fading better than OSSB does in any $\lambda_c$.

## 5. Conclusions

In this paper, the irregular $\theta$ method was used to overcome the dispersion power fading on the RoF link using DD-MZM as an *E/O* converter. The level of dispersion effect on the recovered RF signal power was measured using (*C/N*) deviation factor. Two irregular $\theta$ was set for each *m* used. Results suggested that the irregular $\theta$ overcomes dispersion power fading better than OSSB modulation scheme at all *m* tested. In addition, the RoF link with irregular $\theta$ handled dispersion fading well at *m* ≤ 1, and it has a fixed (*C/N*) deviation factor at all $f_m$ and $\lambda_c$, which is 0.1 at *m* = 1. All in all, the irregular $\theta$ manages to overcome the dispersion power fading at any $f_m$ and $\lambda_c$ without having to re-adjust the transmitter.

**Author Contributions:** Conceptualization, F.U.; methodology, F.U. and T.F.; validation, G.W. and P.S.P.; formal analysis, F.U. and T.F; investigation, F.U. and T.F.; data curation, F.U.; writing—original draft preparation, F.U.; writing—review and editing, G.W. and P.S.P.; supervision, G.W. and P.S.P.; project administration, G.W. and P.S.P.; funding acquisition, G.W.

**Funding:** This research was funded by the Ministry of Research, Technology, and Higher Education of the Republic of Indonesia, under Penelitian Disertasi Doktor (PDD) grant No. NKB-1837/UN2.R3.1/HKP.05.00/2019.

**Conflicts of Interest:** The authors declare no conflict of interest.

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
