# Peer review of "Irregular Shifting of RF Driving Signal Phase to Overcome Dispersion Power Fading"

_photonics, doi:10.3390/photonics6040104_

Round 1

Reviewer 1 Report

The paper presents a theoretical study and simulated results about irregular RF shifting on Dual-Drive Mach-Zehnder modulator (DD-MZM) to lessen fiber dispersion-induced power fading in radio-over-fiber (RoF) systems. Authors experience various parameters that affect the quality of the heterodyne signals. After the authors improve some aspects below, I recommend to publish this work to Photonics MDPI. The following should be addressed.

1. Pg 1, line 28: E/O is usually employed in the central office (CO). Authors may change the base station (BS) by central office (CO).

2. Pg 2: Authors present the introduction section in a very good way. They demonstrate dispersion power fading using a single-mode laser with external intensity modulation and some solutions to overcome this effect. In order to be a comprehensive introduction, authors should add few sentences about the dispersion power fading based on optical frequency comb (OFC) sources such as mode-locked lasers (MLLs) with proposed solutions. The following references are related;
a. H. H. Elwan et al., “Simplified chromatic dispersion model applied to ultra-wide optical spectra for 60 GHz radio-over-fiber systems”, IEEE J. Lightw. Technol., 10.1109/JLT.2019.2929355, July 2019.
b. F. Brendel, et al., “Chromatic dispersion in 60 GHz radio-over-fiber networks based on mode-lock lasers,” IEEE J. Lightw. Technol., vol. 29, no. 24, pp. 3810-3816, Dec. 2011.
c. H. Rzaigui, et al., ”Optical heterodynin for reduction of chromatic dispersion sensitivity in 60 GHz mode-locked lasers systems” IEEE J. Lightw. Technol., vol. 31, no. 17, pp. 2955-2960, Sept. 2013.

3. Pg 3, line 97: It is better to put Eq. (2) before Eq. (1) with modifying the sentence in line 97.

4. Pg 4, line 123: why do you put prime symbol on 2!! should be removed.

5. Pg 6, line 151: Due to my mathematical derivations, I think that (Ac) term should influence in Eq. (23). Could you please recheck and show the derivation of this equation.

6. Pg 7, Figs. 4: In the simulation results, how do you calculate the power without fiber (Prec(L)without fiber) for C/Npenalty and add that in the paper?

7. Pg 7, lines. 202: On the ROF link with variation fm, the bigger the fm used, the closer the distance of the deep fade. Conversely, on the ROF link with variation λc, smaller λc results in a closer distance of the deep fade. It can be noticed on Fig. 5 (b) that BIGGER λc results in a closer distance of the deep fade. Could you please clarify this point?

8. Pg 9, line 233: You already mention this sentence in the same page (line 230), so you can remove the line 233.

9. Pg 11, line 277: Please remove “in other words”.

10. Pg 13, line 322: Alter 3.3 km by 3.2 km to match with the same length you mentioned before (Pg. 7 line 187).

11. Pg 13, Fig. 11: the curves are not clear, could you please distinguish them (increasing the thickness of curve…)

12. Pg 14, line 344: Put space between 40 and GHz.

13. Usually we use (RoF) instead of (ROF).
Also, you make abbreviation for quadrature bias point (QBP) in Pg. 2, line 64, and you repeat the same thing in Pg. 4 line 120. So you can use QBP directly in Pg. 4… could you please check the
other abbreviations?

Reviewer 2 Report

The presented technique to increase the robustness of optical links against the chromatic dispersion (CD) seems effective. The adopted model is convincing, showing how an irregular setting of the phase difference between the arms of a dual-drive Mach Zehnder modulator (DD-MZM), depending on the employed modulation index, can dramatically reduce the sensitivity to CD at the receiver. Nonetheless, in the Reviewer's opinion, the work can be improved before publication.

First of all, the Authors completely overlook mentioning the fact that CD is a perfectly linear effect, which can be mitigated by spools of CD-compensation fibers, properly inserted in the optical fiber links. The Authors should motivate why the solution they propose is to be preferred to CD-compensatiion fiber.

The Authors build their work around the concept of C/N... without explaining not even once that the acronym stands for carrier-to-noise ratio. The Reviewer suggests to specify it in the text, for the sake of clarity.

In Eq. (20), n is the index of the harmonics of the modulation, whereas in (27) it represents the number of samples. Please, resolve this ambiguity.

Why do the Authors choose to consider the sidebands up to the 10th order? Is it really necessary, when usually after the 3rd can be neglected? Please motivate.

In table 2, why the ER of the MZM is considered 60 db? It is way too much, compared to real devices.

In table 3, REference wavelength is 4 V.

Some typos have been detected, like in line 320 "same performance in handles", Figure 8 is mentioned as Gambar 8, in line 347 "better tha in any fm".
